

# Downregulated miRNA-491-3p accelerates colorectal cancer growth by increasing uMtCK expression

Xingkui Tang[1,*], Yukun Lin[2,*], Jialin He[1], Xijun Luo[1], Junjie Liang[1] and Xianjun Zhu[1]

[1] Department of General Surgery, Panyu District Central Hospital, Guangzhou, China
[2] Department of Electron Microscopy, Zhongshan School of Medicine, Sun Yat-sen University, Guangzhou, China
[*] These authors contributed equally to this work.

## ABSTRACT

Colorectal carcinoma (CRC) is the second most frequent cancer worldwide. MiR-491-3p, a tumor-suppressive microRNA (miRNA, miR), has been revealed to be abnormally expressed in CRC tissues. Meanwhile, up-regulated ubiquitous mitochondrial creatine kinase (uMtCK) contributes to CRC cell proliferation. Here we aim to explore whether aberrant miR-491-3p expression promotes CRC progression through regulating uMtCK. To this end, miR-491-3p and uMtCK levels were assessed in CRC tissues using quantitative real-time PCR (qRT-PCR). The biological roles of miR-491-3p and uMtCK in regulating CRC growth were evaluated using colony formation assay and mouse Xenograft tumour model. We found that miR-491-3p expression was decreased in CRC tissues compared with matched para-cancerous tissues, whereas uMtCK expression was increased. Functionally, miR-491-3p overexpression repressed SW480 cell growth, whereas miR-491-3p depletion accelerated SW620 cell proliferation and growth. Inversely, uMtCK positively regulated CRC cell proliferation. Mechanistically, miR-491-3p post-transcriptionally downregulated uMtCK expression by binding to 3'-UTR of *uMtCK*. Consequently, restoring uMtCK expression markedly eliminated the role of miR-491-3p in suppressing CRC growth. Collectively, miR-491-3p functions as a tumour suppressor gene by repressing uMtCK, and may be a potential target for CRC treatment.

Corresponding authors
Xingkui Tang, tangxingkui@163.com
Yukun Lin, linyukun@mail.sysu.edu.cn

## INTRODUCTION

CRC is the second most frequent malignancy in the western world (*Mano & Humblet, 2008*), and is the sixth most common cancer in China (*Siegel et al., 2017*), with an approximated 1.5 million new cases and 53, 000 new deaths worldwide in 2021 (*Siegel et al., 2021*). Surgical excision is the mainstay treatment for CRC (*Galli & Rosenberg, 2018*). Mounting evidence has demonstrated that neoadjuvant chemoradiotherapy (nCRT) has significantly reduced relapse rate in locally advanced colon cancer (LACC) patients and increased its survival rate (*Dienstmann, Salazar & Tabernero, 2015*; *Yin et al., 2021*).

However, the major reason for cancer-related mortality is caused by distal metastasis and chemoradiotherapy resistance (*Van Cutsem et al., 2009*).

MiRNA is a kind of small non-coding RNA transcript consisting of 19–25 bases (*Goldberger et al., 2013*; *Berindan-Neagoe et al., 2014*; *Boissinot et al., 2020*); it plays key roles in many physiopathological processes, such as energy metabolism, tumorigenesis, and immune response (*Lenart et al., 2020*; *Gangaraju & Lin, 2009*; *Taniguchi, Uchiyama & Akao, 2021*). Mounting studies have revealed that a lot of miRNAs are dysregulated in CRC (*Shi, Zhou & Zhang, 2021a*; *Lan et al., 2021*; *Zhang et al., 2021*), prostate cancer (*Coarfa et al., 2016*), and breast cancer (*Lorenzo-Martin et al., 2019*), etc. MiRNAs post-transcriptionally regulate gene expression by binding to the 3′ untranslated region (3′ UTR) of target mRNAs, making possible to function as oncogenes, tumor suppressors, or drug resistance factors in different cases (*Shi, Zhou & Zhang, 2021a*; *Lan et al., 2021*; *Zhang et al., 2021*; *Wang et al., 2021*). Dysregulated expression of miRNAs is correlated with cancer cell proliferation, migration, and invasion (*Zhang et al., 2021*; *Li et al., 2015*; *Duan et al., 2017*; *Lan et al., 2021*; *Ding et al., 2022*). A high-throughput sequencing study has revealed that more than 150 miRNAs are dysregulated in CRC tissues, including 84 upregulated and 70 downregulated (*Hozaka et al., 2021*). MiR-491-3p is one of the mature products of miR-491. Recently, miR-491-3p has been reported to be involved in the progression of several cancers, such as retinoblastoma, tongue cancer, glioblastoma, and gastric cancer (*Hu et al., 2021*; *Zheng et al., 2015*; *Li et al., 2015*; *Yu & Luo, 2022*). Furthermore, it has been demonstrated that miR-491 facilitates CRC cell apoptosis by reducing Bcl-XL expression (*Nakano et al., 2010*). However, whether and how miR-491-3p involves in CRC tumorigenesis remains to be explored.

Ubiquitous mitochondrial creatine kinase (uMtCK) is one of the creatine kinases, which play a key role in mitochondrial energy metabolism. uMtCK functions as an oncogene in pathological processes of several types of cancer including breast cancer (*Qian et al., 2012*), hepatocellular carcinoma (*Uranbileg et al., 2014*), and CRC (*Zhang et al., 2019*). A recent study showed that miR-519b-3p suppresses CRC cell proliferation and migration through decreasing uMtCK expression (*Zhang et al., 2019*). A bioinformatics analysis revealed the presence of binding site of miR-491-3p in uMtCK-3′ UTR, indicating that miR-491-3p may regulate the biological behavior of CRC cells through targeting uMtCK.

Based on the above findings, the biological role of miR-491-3p and uMtCK in CRC progression was investigated. We demonstrated that miR-491-3p acted as a tumour suppressor in CRC, whereas uMtCK functioned as an oncogene. More important, overexpression of miR-491-3p supressed CRC progression by post-transcriptionally repressing uMtCK expression.

## MATERIALS AND METHODS

### Clinical specimens

The study was approved by the Ethics Committee of Panyu Central Hospital (PYRC-2021-113). Twenty-one pairs of colon cancer tissues and matched para-cancerous tissues (5 cm from tumor edges) (*Qureshi et al., 1994*; *Jiang et al., 2020*) were obtained from Panyu

Central Hospital (8 females, 13 males; age range: 23.8–88.1 years; stage: II-III). Pathological diagnosis was performed by two skilled pathologist. Informed consent was obtained from all donors before biopsies.

## Cell culture

Three CRC cell lines, SW480, SW620, and HCT116, were purchased ATCC (Manassas, USA). A normal human colon mucosal epithelial cell line, NCM460, was obtained from Fenghui Biotechnology (Hunan, China). These cells were cultured in DMEM medium (Gibco, Carlsbad, CA, USA) containing 10% FBS (Absin, Shanghai, China) and 1% penicillin/streptomycin (ThermoFisher Scientific, Waltham, MA, USA), in a humidified atmosphere with 5% $CO_2$ at 37 °C.

## qRT-PCR

Total RNA was extracted from tissues and cells with Trizol reagent (Catalog Number: 9108; Takara, Shiga, Japan) in accordance with manufacturer's instructions. SuperScript IV reverse transcriptase (ThermoFisher Scientific, Waltham, MA, USA) was used to synthesize first-strand cDNAs at 53 °C for 10 min. qRT-PCR was carried out in triplicate on MA-6000 quantitative PCR system (Molarray, Jiangsu, China) with Brilliant II qPCR reagent (Agilent, Santa Clara, CA, USA). miR-491-3p level was normalized by U6, and uMtCK level was normalized by GAPDH. Primer sequences used for qRT-PCR were listed in Table 1.

## Overexpression and RNA interference (RNAi)

Recombined plasmid expressing uMtCK (pcDNA-uMtCK) was generated by cloning complete codon sequence of uMtCK into pcDNA 3.1 vector. Unmodified pcDNA 3.1 plasmid was used as control (pcDNA3.1-Cont). Small interfering RNAs (siRNAs) against uMtCK (siuMtCK) and negative control (siCont) were obtained from Sangon (Shanghai, China). The siuMtCK sequence was 5′-UUCACAAUCAAUCAAAUAGUUCUAUUUGAUUGAUUGUGAACG-3′and the siCont 5′-UUAAAUGUGAGCGAGUAACAAGUUACUCGCUCACAUUUAAUG-3′. MiR-491-3p mimic, control miRNA mimic, miR-491-3p inhibitor, and negative control miRNA inhibitor (N.C.) were purchased from Genewiz (Jiangsu, China). Indicated plasmids, miRNA mimics, miRNA inhibitors, and siRNAs were transfected into cells with Lipofectamine 3000 (Invitrogen, Carlsbad, CA, USA). The expression level after transfection were assayed using qRT-PCR.

## Cell proliferation

Cell proliferation was assessed using CCK-8 reagent (Yeasen Biotechnology, Shanghai, China) in accordance with the manufacture' s protocol. SW480 or SW620 cells were seeded into 96-well plates (5,000 cells/well). After incubating for 24 h, SW480 cells were transfected with miR-491-3p mimic or control mimic, and SW620 cells were treated with miR-491-3p inhibitor or control miRNA inhibitor, as mentioned earlier. Cell proliferation was assayed after transfection for indicated time durations, by adding CCK-8 reagent (10 μL) into per well. After 60 min of incubation, absorbance of each well was assessed on a HBS- ScanX microplate reader (DeTieLab, Jiangsu, China).
**Table 1** Primer sequences for qRT-PCR.

| Gene name | Primers sequences (5′–3′) |
| --- | --- |
| GAPDH | Sense: 5′- GGAGAAACCTGCCAAGTATGA-3′<br>Antisense: 5′-TCCTCAGTGTAGCCCAAGA-3′ |
| uMtCk | Sense: 5′- AAGCGTGGTACTGGAGGAGT -3′<br>Antisense: 5′- AGCTCCACCTCTGATTTGCC -3′ |
| U6 | Sense: 5′- CTCGCTTCGGCAGCACA-3′<br>Antisense: 5′- CGCTTCACGAATTTGCGTGTC-3′ |
| miR-491-3p | RT: 5′- GTCGTATCCAGTGCAGGGTCCGAGGTATTCG<br>CACTGGATACGACGTAGAA-3′<br>Sense: 5′- GCGCTTATGCAAGATTCC-3′<br>Antisense: 5′- AGTGCAGGGTCCGAGGTATT-3′ |

## Colony formation assay

SW480 cells were treated with miR-491-3p mimic, control miRNA mimic or both miR-491-3p mimic and pcDNA-uMtCK and seeded (200 cells per well) into six-well plates. SW620 cells were treated with miR-491-3p inhibitor and seed as mentioned above. Next, cells were cultured at 37 °C for 14 days. After rinsing with PBS three times, colonies were fixed by 4% PFA and dyed using 0.2% crystal violet. Colonies of >50 μm diameter were counted by Fiji tool (NIH, Bethesda, MD).

## Dual luciferase reporter gene assay

Recombinant plasmids of pGL3-uMtCK-3′ UTR-wt (uMtCK-3′ UTR) or its mutant (uMtCK-3′ UTR-mut) were constructed by cloning 3′ UTR of uMtCK or its mutant into pGL3 vector (Fenghui Biotechnology). SW480 cells (50,000 cells/well) were seeded into 48-well plate, and pGL3-uMtCK-3′ UTR-wt or uMtCK-3′ UTR-mut was co-transfected with 20nM of miR-491-3p and 2 ng of pRL-TK into SW480 cells. After transfecting for 48 h, luciferase activity was examined using Pierce™ Renilla-Firefly Luciferase Dual Assay Kit (ThermoFisher Scientific, Waltham, MA, USA). Transfection efficiencies were normalized by renilla luciferase.

## Western blot

Cells were homogenized and total protein was obtained using RIPA buffer (Abcam, Cambridge, UK) containing protease inhibitor mixture (ThermoFisher Scientific, Waltham, MA, USA). Protein concentration was assessed with BCA Protein Assay Kit (Sangon, Shangai, China). Fifty μg of protein of each lysate was electrophoresed through 10% SDS-PAGE and then transferred to PVDF membrane (Absin, Shanghai, China). Membranes were blocked with 2% BSA (Yeasen, Shanghai, China) and then labeled with primary antibodies against uMtCK (1:1000, PA5-96224; ThermoFisher Scientific, Waltham, MA, USA) and $\beta$-actin (1:2000, ab8226; Abcam, Cambridge, UK) at 37 °C for 90 min. Membranes were subsequently rinsed with PBST, and then incubated with HRP-conjured secondary antibody (1:1000; Beyotime, Jiangsu, China). Immunoblots were visualized using ECL luminescence reagent (Absin, Shanghai, China).

## Xenograft mouse model

Athymic BALB/c mice (4-6 weeks old) were obtained from Charles River (Beijing, China) and maintained under pathogen-free conditions. All mice were raised at a 12 h dark/light cycle, at $24 \pm 2$ °C, and were given free access to food and water. Animal experiments were carried out in approval of the Experimental Animal Committee of Panyu District Central Hospital (PYRC-2021-113), and conducted under the 3R principle to reduce suffering of mice. SW480 cells ($5 \times 10^6$) treated with miR-491-3p mimic, control miRNA mimic or both miR-491-3p mimic and pcDNA-uMtCK were resuspended in 100 µL of PBS and injected into right flank of mice ($n = 3$). Tumor growth was half-quantified with a caliper in a blinded fashion at the indicated time (0, 14, 21, and 35 days). Tumor volume was estimated through the formula length $\times$ width$^2$ $\times 0.5$, as previously described (*Hart et al., 2012*; *Khare et al., 2019*). At the end of studies, all mice were euthanatized by cervical dislocation under inhalation anesthesia (2% isoflurane), and tumor tissues were collected. During the experiment, once any mouse suffering from unexpected disease or over-sized tumor (>1 cm) will be euthanatized as described earlier.

## Data analysis

All values were showed as mean $\pm$ standard error from three independent experiments. Statistical analysis between two groups was carried out using student's $t$-test, and comparisons among multiple groups were performed using one-way analysis of variance (ANOVA) followed by the Scheffé test (GraphPad Prism, San Diego, CA, USA). The value of p less than 0.05 was considered significant.

# RESULTS

## MiR-491-3p and uMtCK expression in CRC tissues

To investigate the biological role of miR-491-3p in CRC, miR-491-3p expression was first assessed in CRC tissues and cell lines. Figs. 1A and 1B showed that miR-491-3p level was markedly down-regulated in CRC specimens and cell lines. On the contrary, uMtCK expression was prominently increased in CRC specimens when compared to matched para-cancerous tissues (Fig. 1C). uMtCK expression was also increased in CRC cells (SW480, SW620, and HCT116) compared with NCM460 (Fig. 1D). We further analyzed uMtCK level in 275 colon cancer (COAD) specimens and 349 controls from the TCGA/GTEx COAD dataset, and in 92 rectal cancer (READ) and 318 normal tissues from the TCGA/GTEx READ dataset using GEPIA tool (http://gepia2.cancer-pku.cn/). Figure 1E revealed that uMtCK level was significantly up-regulated in COAD and READ specimens when compared to normal specimens. Additional, the miR-491-3p level was negatively associated with the uMtCK level in 21 CRC specimens (Fig. 1F).

## The role of miR-491-3p and uMtCK in regulating CRC cell proliferation and growth

Then the biological effect of miR-491-3p on regulating CRC cell proliferation and growth was explored. Given that the endogenous miR-491-3p level was the lowest in SW480 cells and was the highest in SW620 cells in CRC cells (Fig. 1B), miR-491-3p overexpression was

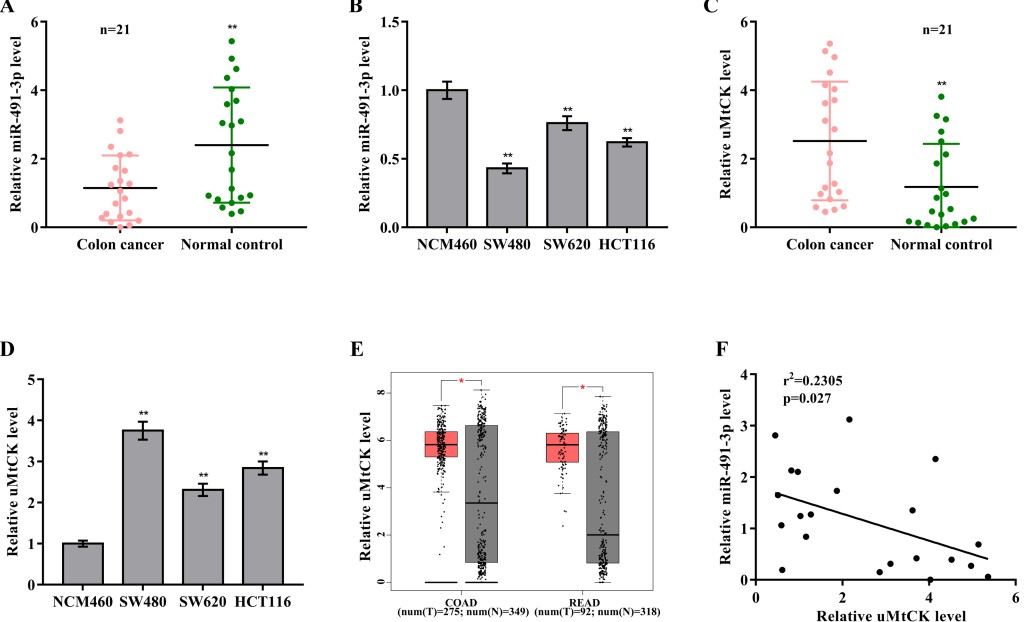

**Figure 1** **MiR-491-3p and uMtCK expression in CRC tissues.** (A) qRT-PCR analysis was applied to assess miR-491-3p expression in colon cancer specimens and matched normal tissues ($n = 21$, $p < 0.01$). (B) qRT-PCR analysis was applied to assess miR-491-3p level in NCM460, SW480, SW620 and HCT116 cells. (C) qRT-PCR analysis of uMtCK level in colon cancer specimens and matched normal tissues ($n = 21$, $p < 0.01$). (D) qRT-PCR analysis was carried out to assess uMtCK expression in NCM460, SW480, SW620 and HCT116 cells. (E) uMtCK expression was analysed in COAD and READ from TCGA database was analysed through GEPIA tool. Tumour samples were showed as red and normal samples were showed as grey. (F) The correlation of miR-491-3p level with uMtCK level was analysed in 21 colon cancer tissues. $^{*} p < 0.05$, $^{**} p < 0.01$.

carried out in SW480 cells (Fig. 2A) and miR-491-3p knockdown was carried out in SW620 cells (Fig. 2E), respectively. Forced expression of miR-491-3p repressed cell proliferation (Fig. 2B) and growth (Fig. 2C and 2D), whereas miR-491-3p knockdown accelerated cell proliferation (Fig. 2F) and growth (Fig. 2G and 2H). In an addition, uMtCK overexpression accelerated cell proliferation (Figs. 3A and 3B), whereas uMtCK depletion significantly decreased cell proliferation (Figs. 3C and 3D). These data demonstrate that miR-491-3p can exert a function of tumour suppressor, whereas uMtCK exerts an oncogene in CRC.

## MiR-491-3p post-transcriptionally downregulated uMtCK expression
The regulatory effect of miR-491-3p on uMtCK expression was next explored. Bioinformatics analysis from TargetScan showed that miR-491-3p directly targets 3′ UTR of *uMtCK* gene (Fig. 4A). To affirm the direct combination of miR-491-3p with position 17-23 of *uMtCK* 3′ UTR, the recombinant plasmids of pGL3-uMtCK-3′ UTR-wt (uMtCK-3′ UTR) or its mutant (uMtCK-3′ UTR-mut) were transfected into SW480 cells together with miR-491-3p. Figure 4B showed that miR-491-3p markedly repressed the luciferase activity of uMtCK-3′ UTR compared with control, whereas mutation of four nucleotides in uMtCK-3′ UTR caused a complete abrogation of the suppressive effect. Although

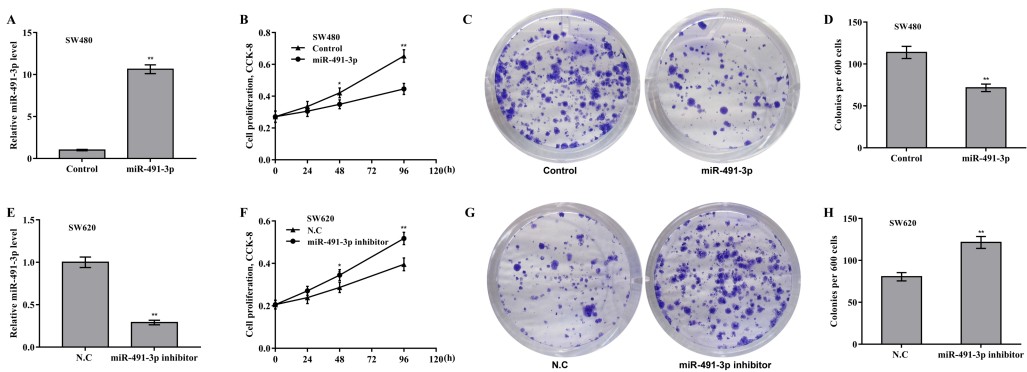

**Figure 2 MiR-491-3p repressed CRC cell proliferation and growth.** (A) SW480 cells were treated with miR-491-3p mimics for 48 h and then qRT-PCR analysis was carried out to assess miR-491-3p level. (B) CCK-8 was applied to assess SW480 cell proliferation after miR-491-3p overexpression at the indicated time points. Colony formation assay (C) and quantitative analysis (D) was carried out to assess SW480 cell growth after miR-491-3p overexpression for 14 days. (E) SW620 cells were treated with miR-491-3p inhibitor for 48 h and then qRT-PCR analysis was applied to assess miR-491-3p level. (F) CCK-8 was applied to assess SW620 cell proliferation after miR-491-3p inhibition at the indicated time points. Colony formation assay (G) and quantitative analysis (H) was carried out to assess SW620 cell growth after miR-491-3p inhibition for 14 days. * $p < 0.05$, ** $p < 0.01$.

miR-491-3p did not change the mRNA level of uMtCK in CRC cells (Fig. S1), miR-491-3p overexpression markedly decreased uMtCK protein expression (Figs. 4C and 4D).

## MiR-491-3p repressed CRC growth by repressing uMtCK expression

Finally, we investigated whether miR-491-3p negatively regulated CRC cell proliferation and growth by inhibiting uMtCK expression. Figures 5A and 5B revealed that forced expression of miR-491-3p repressed SW480 cell proliferation and growth, whereas restoring uMtCK expression by transfecting pcDNA-uMtCK prominently decreased the role of miR-491-3p in suppressing cell proliferation and growth. More important, miR-491-3p repressed the growth of CRC xenografts in nude mice, whereas re-expression of uMtCK significantly damaged the effect of miR-491-3p on suppressing tumour growth *in vivo* (Figs. 5C and 5D). These data demonstrate that miR-491-3p represses CRC cell proliferation and growth, at least in part by post-transcriptionally inhibiting uMtCK expression.

## DISCUSSION

MiR-491-3p exerts a tumour suppressor effect on many human malignancies, such as retinoblastoma (*Hu et al., 2021*), osteosarcoma (*Duan et al., 2017*), hepatocellular carcinoma (HCC) (*Zhao et al., 2017*), and glioblastoma (*Li et al., 2015*). MiR-491-3p expression is also decreased in CRC tissues and oxaliplatin-resistant CRC cells (*Guo et al., 2017*; *Tang et al., 2019*). However, the biological role of miR-491-3p in CRC cell proliferation remains poorly understood. Here, we demonstrated that, (i) MiR-491-3p expression was downregulated and uMtCK expression was upregulated in CRC tissues, (ii) MiR-491-3p repressed CRC cell proliferation and growth through inhibiting uMtCK
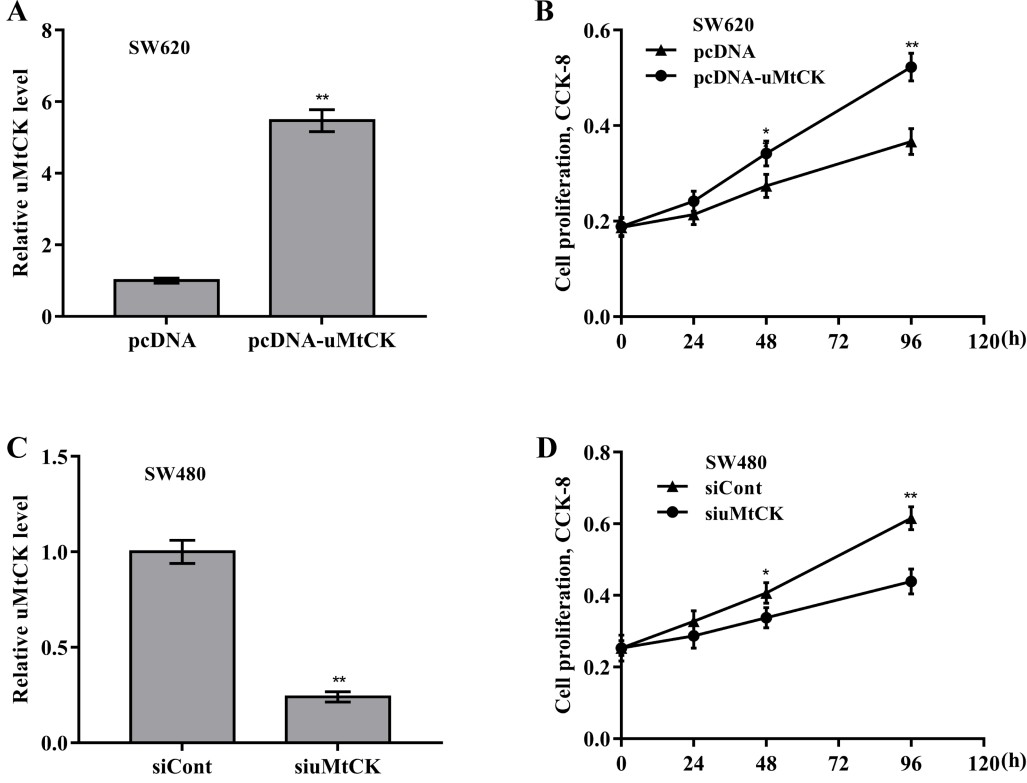

**Figure 3** **uMtCK accelerated CRC cell proliferation and growth.** (A) SW620 cells were treated with recombinant pcDNA-uMtCK plasmids and then uMtCK level was assessed using qRT-PCR 48 h later. (B) CCK-8 was applied to assess SW620 cell proliferation after uMtCK overexpression at the indicated time points. (C) SW480 cells were treated with siuMtCKs for 48 h and then uMtCK level was assessed using qRT-PCR. (D) CCK-8 was applied to assess SW480 cell proliferation after uMtCK knockdown at the indicated time points. $* p < 0.05$, $** p < 0.01$.

expression. These data identified the role of miR-491-3p/uMtCK axis in CRC progression and represent a prospective opportunity to treat CRC.

Creatine kinases are crucial to energy metabolism, especially in tissues that require high energy such as brain, placenta, and myocardium (*Schlattner, Tokarska-Schlattner & Wallimann, 2006*; *Whittington et al., 2018*). uMtCK mainly localized in mitochondria, exerts a central regulatory role in energy homeostasis (*Qian et al., 2012*). Rapid proliferation of tumour cells requires more energy than normal cells (*Cheng et al., 2019*). Based on the above analysis we hypothesized that aberrant expression of uMtCK might frequently occur in tumour tissues. Indeed, emerging studies have revealed the overexpression of uMtCK in many human tumours including breast cancer (*Qian et al., 2012*), HCC (*Uranbileg et al., 2014*), and CRC (*Zhang et al., 2019*). *Zhang et al. (2019)* showed that uMtCK expression is remarkably increased in CRC tissues, and miR-519b-3p-mediated inhibition of uMtCK causes a significant decrease in CRC cell proliferation and invasion. Upregulated uMtCK is significantly associated with shorter survival in breast cancer patients (*Qian et al., 2012*). uMtCK accelerates cancer cell proliferation through regulating mitochondrial apoptotic

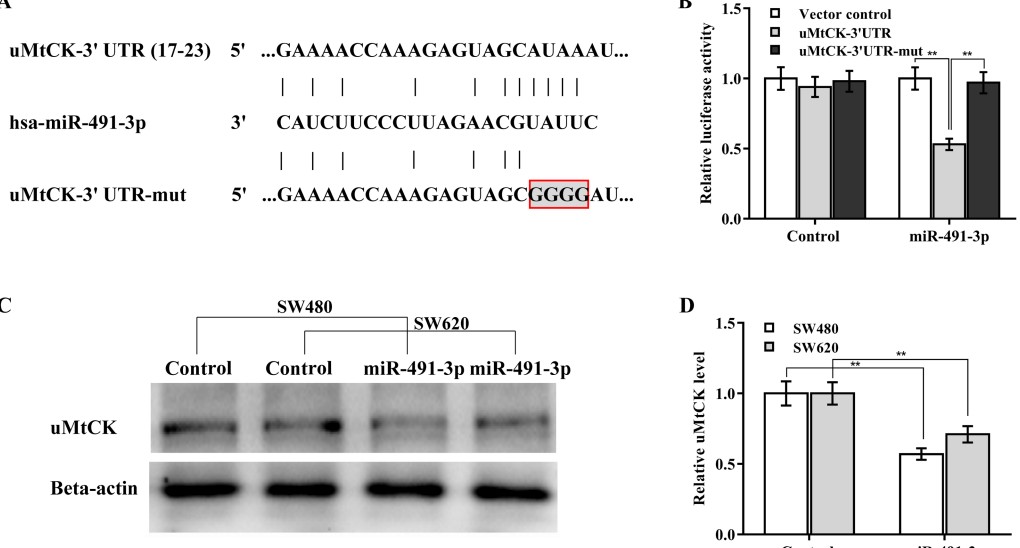

**Figure 4** **MiR-491-3p post-transcriptionally repressed uMtCK expression.** (A) Schematic illustration of the predicted binding site of miR-491-3p with uMtCK. (B) pGL3-uMtCK-3′ UTR-wt (uMtCK-3′ UTR) or its mutant (uMtCK-3′ UTR-mut) were co-transfected with miR-491-3p into SW480 cells, and dual luciferase reporter gene assay was applied to measure luciferase activity. Western blot (C) and quantitative analysis (D) of uMtCK expression in CRC cells after miR-491-3p overexpression. ** $p < 0.01$.

pathway (*Qian et al., 2012*). However, several studies converge on the opposite conclusion. *Shi et al. (2021b)* demonstrated that uMtCK level is decreased in lower-grade glioma tissues (LGG), and that lower expression of uMtCK is correlated with a worse prognosis. uMtCK expression is reported to be decreased in prostate cancer progresses and may be correlated with a metabolic switch in ATP usage (*Amamoto et al., 2016*). These studies suggest that uMtCK plays different roles in different types of tumor. However, the mechanism underlying uMtCK involvement in cancer progression remains unknown, and the role of uMtCK in CRC is still unclear.

In the study we demonstrated that uMtCK expression is prominently increased in CRC tissues and cell lines. uMtCK expression was analysed in COAD and READ tissues from the TCGA/GTEx dataset. These data validated that uMtCK expression is prominently increased in CRC tissues. uMtCK overexpression facilitates CRC cell proliferation, whereas uMtCK depletion represses CRC cell proliferation. Then, the mechanism underlying uMtCK overexpression was explored. A downregulated miRNA in CRC, miR-491-3p, is responsible for increasing uMtCK expression. Forced expression of miR-491-3p could markedly repress the luciferase activity of uMtCK-3′ UTR. Although miR-491-3p could not change uMtCK mRNA level in CRC cells, miR-491-3p overexpression inhibits uMtCK protein expression, indicating that miR-491-3p post- transcriptionally downregulates uMtCK expression in CRC cells. Functionally, miR-491-3p overexpression represses CRC cell proliferation, whereas restoring uMtCK expression partially eliminates the role of

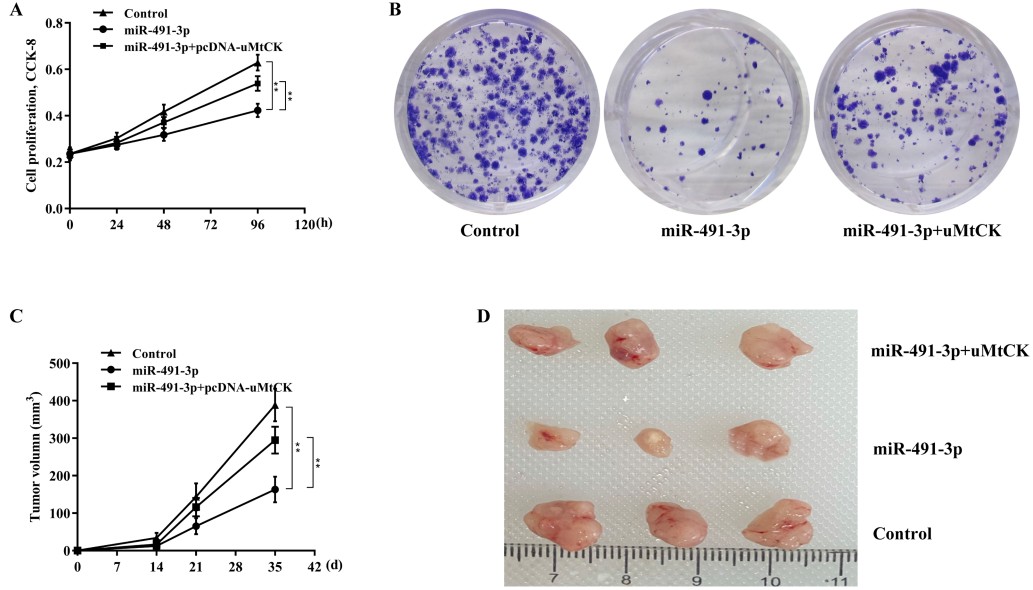

**Figure 5** **MiR-491-3p repressed CRC growth by repressing uMtCK expression.** (A) CCK-8 was applied to assess SW480 cell proliferation after miR-491-3p overexpression in the presence or absence of uMtCK overexpression. (B) Colony formation assay was applied to assess SW480 cell growth after miR-491-3p overexpression in the presence or absence of uMtCK overexpression for 14 days. (C) SW480 cells overexpressed with miR-491-3p and uMtCK were subcutaneously injected into dorsal flanks of nude mice and tumour growth was calculated at different time points (0, 14, 21, and 35 days). (D) Tumour tissues in different groups at day 35 were showed. ** $p < 0.01$.

miR-491-3p in inhibiting cell growth *in vitro* and *in vivo*. In conclusion, downregulated miRNA-491-3p accelerates colorectal cancer growth by increasing uMtCK expression.

### Funding

This work was supported by the Science and Technology Project of Guangzhou (No. 201904010070). The funders had no role in study design, data collection and analysis, decision to publish, or preparation of the manuscript.

### Grant Disclosures

The following grant information was disclosed by the authors:
Science and Technology Project of Guangzhou: 201904010070.

### Competing Interests

The authors declare there are no competing interests.

### Author Contributions

• Xingkui Tang conceived and designed the experiments, analyzed the data, prepared figures and/or tables, authored or reviewed drafts of the article, and approved the final draft.
- Yukun Lin analyzed the data, prepared figures and/or tables, authored or reviewed drafts of the article, and approved the final draft.
- Jialin He conceived and designed the experiments, performed the experiments, authored or reviewed drafts of the article, and approved the final draft.
- Xijun Luo performed the experiments, authored or reviewed drafts of the article, and approved the final draft.
- Junjie Liang analyzed the data, authored or reviewed drafts of the article, and approved the final draft.
- Xianjun Zhu performed the experiments, authored or reviewed drafts of the article, and approved the final draft.

### Animal Ethics

The following information was supplied relating to ethical approvals (i.e., approving body and any reference numbers):

The Ethics Committee of Panyu Central Hospital approved this study (PYRC-2021-113).

### Data Availability

The raw data is available in the Supplemental Files.

### Supplemental Information

Supplemental information for this article can be found online at http://dx.doi.org/10.7717/peerj.14285#supplemental-information.

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
