# Peer review of "Downregulated miRNA-491-3p accelerates colorectal cancer growth by increasing uMtCK expression"

_PeerJ, doi:10.7717/peerj.14285_

## Round 0.1 · original submission · Major Revisions

Please answer the reviewer's comments carefully.

Reviewer 1 has requested that you cite specific references. You are welcome to add it/them if you believe they are relevant. However, you are not required to include these citations, and if you do not include them, this will not influence my decision.

Reviewer 1 ·

Basic reporting

Tang used qRT-PCR to detect the expression of miR-491-3p and uMtCK in CRC tissues and use colony formation assay and mouse Xenograft tumour model to asses the biological role of miR-491-3p and uMtCK .The quality of the manuscript is general, and there are a couple of issues that need to be addressed before further consideration

Experimental design

Do normal tissues come from healthy controls or paracancerous tissues? Please explain clearly

I recommend author use multiple databases such asmiRDB, starBase and miRTarBase to predicted target mRNAs of microRNA-491-3p and construct a Venn plot.They can refer to PMID: 35615948

Validity of the findings

There are little references in this study, please add some related references. (recommend PMID: 35815280 and PMID: 35432591).

If the author wants to upload Figure1 raw data as Supplemental files, please rearrange the Figure in it
A conclusion figure (graphical abstract) will be very useful for the readers

The grammar and the subject used in the manuscript need to be comprehensively corrected and improved by a fluent speaker.

Reviewer 2 ·

Basic reporting

This is an interesting study which interrogated whether the miR-491-3p could repress cancer proliferation via inhibiting uMtCK expression. However, there are many points need to be addressed before considering any further decision. The English writing needs to be improved.

Experimental design

Investigation was performed to moderare technical & ethical standard.
Methods description is not sufficient enough.

Validity of the findings

Impact and novelty is at a moderate level.

Additional comments

Please see the attached comments.

Annotated reviews are not available for download in order to protect the identity of reviewers who chose to remain anonymous.

·

Basic reporting

Tang et al. present a manuscript in which they study the role of the negative regulation of miRNA-491-3p in the progression of colorectal cancer, linking this effect to the regulation of uMtCK protein expression. The work carried out by the authors is satisfactory and the relationship between this miRNA and the protein it regulates is clearly justified, so that a new pathway of tumor progression in colorectal cancer is reported.

The text is correctly written and the level of English used is adequate. The introduction presented by the authors helps the reader to be informed about the state of the art of colorectal cancer and the importance of miRNAs such as the one studied in the development and progression of CRC, providing updated references during the writing process. However, I suggest that from line 77 onwards the authors relate the objectives sought in this research together with their hypothesis, leaving the presentation of results for later.

The results shown by the authors are consistent with the hypothesis put forward and the level of quality of these is correct. I consider that the authors could include the gene expression results on the variation of uMtCK levels by miR-491 (they indicate that these data are not visible in line 197). In the results of the in vivo experiment, the number of mice that were included in each study group should be named. This information could be indicated in the text and in the caption of the respective figure (5C).

Talking about the graphs, I would like the authors to modify the layout of Figure 4C to simplify it for the reader and to modify the legends of Figures 5A and 5C so that they do not overlap with the respective cell proliferation and tumor volume graphs. On the other hand, it would be good to indicate on the X-axis of all cytotoxicity plots that what is shown is the optical density value of CCK8.

Finally, the discussion of results by the authors is correct, relating the results of several previous works performed in other types of cancer mainly with those obtained by them. However, I think that the description of results made in lines 215-219 could be better integrated during the writing of the section to increase its quality. Therefore, I think they could try to rewrite parts of this complete section to increase cohesion. Finally, the conclusion should be integrated into the text and be more explanatory, perhaps by extending it a bit.

Some small details to correct specifically in the article:

1. The authors should correct small typos or expressions in the text. For example, in line 49, to avoid being redundant, I would change "survival rate of LACC patients" to "its survival rate". In line 60, "make it possible" should be replaced by "making possible". On line 123, "and" should be inserted before "were seeded". On line 139, it should be indicated that the value 10 is a percentage. On line 203, there is a typo in "growth".

2. In line 178 (section title), the authors could mention uMtCK, since part of the results described are related to this protein.

3. The authors should review some of the references in the bibliography. Specifically, the reviewer has detected errors in Galli et al. (2018) and Zhang et al. (2021).

Experimental design

The thread that the experiments of this research have is correctly carried out, possessing a high level of quality.
At this point, the reviewer raises the following improvements/questions:

1. Regarding the clinical samples obtained (section 2.1), is information known about the tumor stage of the patients from which they come? This information could be decisive in interpreting the results obtained and would even allow a relationship to be made between stage and the expression of miRNA and the studied protein.

2. In section 2.3, the authors should specify the method/kit used for RNA extraction, given that they only indicate that the samples/tissues are preserved in Trizol.

3. In section 2.9, more information is needed about the injection and how the groups of mice were organized. It is imperative that the authors indicate the number of mice per group and in which solvent the cells were injected. On the other hand, are the tumor cells injected into 2 different flanks of each mouse? This experimental model could cause the results obtained in the in vivo experiment to be inaccurate, given that the growth of both tumors could be slowing down the growth of the other.

4. In section 2.10, the authors describe that they use the ANOVA test followed by a Scheffe post-hoc analysis or a t-test for the analysis between two samples. Since statistical comparisons between more than 2 groups are made throughout the article (e.g., in qPCR results), which statistical test was used? Generally, t-tests (or variants thereof, depending on the degree of normality of the samples) are usually used for analysis of two samples, while ANOVA (or variants thereof) are used for analysis of more than two.

Validity of the findings

After reviewing all the information provided, the validity of all the data in this research appears to be correct, although the possibility arises that the statistical analyses carried out by the authors could be more precise by using other different tests.

The study provides a new discovery in this field of study and the work carried out by the authors is correct, having included the experimental controls required to verify that the system works as reported.

---

## Round 0.2 · Minor Revisions

After a minor revision, this manuscript could be accepted.

Reviewer 2 ·

Basic reporting

This is a re-review step, the authors have addressed all my concern, I have no further comments.

Experimental design

This is a re-review step, the authors have addressed all my concern, I have no further comments.

Validity of the findings

This is a re-review step, the authors have addressed all my concern, I have no further comments.

Additional comments

This is a re-review step, the authors have addressed all my concern, I have no further comments.

·

Basic reporting

The authors have corrected the aspects considered by the reviewers and have discussed those that they do not share correctly.

As a minor revision, the conclusion (lines 255-257) should be integrated into the text, following the format: "In conclusion, downregulated miRNA-491-3p accelerates colorectal cancer growth by increasing uMtCK expression".

Once this is corrected, the manuscript possesses the quality to be accepted for publication.

Experimental design

Not applicable

Validity of the findings

Not applicable

---

## Round 0.3 · accepted · Accept

Thank the authors for their careful response and revision.